# Research on Displacement Monitoring of Key Points in Caverns Based on Distributed Fiber Optic Sensing Technology

**DOI:** 10.3390/s25082619

**Published:** 2025-04-21

**Authors:** Jiangdong Wang, Ziming Xiong, Sheng Li, Hao Lu, Minqian Sun, Zhizhong Li, Hao Chen

**Affiliations:** State Key Laboratory of Disaster Prevention & Mitigation of Explosion & Impact, Army Engineering University of PLA, Nanjing 210007, China; lgdwjd@163.com (J.W.); lisheng19980928@163.com (S.L.); lh829829@163.com (H.L.); emmassun@hotmail.com (M.S.); lizz@aeu.edu.cn (Z.L.); ch15965317925@163.com (H.C.)

**Keywords:** fiber optic sensors, displacement monitoring, numerical simulation, fitting relationship

## Abstract

The accurate and real-time monitoring of key-point displacements in cavern structures is crucial for assessing structural safety and stability. However, traditional monitoring methods often fail to meet the high-precision requirements in complex environments. This study explored the potential application of fiber optic sensors in monitoring key-point displacements by leveraging their sensitivity to optical parameters and spectral changes. Through theoretical analysis, a linear relationship model between key-point displacements and circumferential strain was derived and validated via uniaxial compression tests. Further numerical simulations revealed that different material properties and structural characteristics significantly affect the slope and intercept of the fitting curve, establishing a correlation between these factors and the model parameters. The results demonstrated that fiber optic sensors could accurately measure circumferential strain within the elastic range and reliably reflect key-point displacement trends through the linear relationship model. This paper provides a new theoretical basis for the application of fiber optic sensors in structural health monitoring and expands their potential in civil and geotechnical engineering fields, offering scientific support for engineering design optimization and disaster prevention.

## 1. Introduction

With the increasing scale and depth of underground engineering projects, traditional monitoring methods face challenges in terms of accuracy, adaptability, and long-term stability. In recent years, as energy storage technologies have advanced, cavern structures have emerged as prominent spatial forms in these underground projects [1]. Fiber optic sensing technology, with its advantages of distributed measurement and electromagnetic interference resistance, offers a novel approach to structural health monitoring. Research has demonstrated that distributed fiber optic sensing (DFOS) and fiber Bragg grating (FBG) technologies enable the high-precision monitoring of multiple physical parameters, including structural deformation and temperature fields, and have achieved remarkable success in major projects such as the Brenner Base Tunnel. However, DFOS outperforms traditional fiber optic sensing technologies in terms of coverage, interference resistance, multifunctional integration, and long-term reliability, making it more suitable for large-scale applications such as infrastructure health monitoring and industrial process control. It should be noted, however, that the cost of using DFOS is relatively high and that errors in temperature measurement need to be taken into account. By measuring the optical path difference of combined light signals after passing through a coupler, this technology enables the real-time monitoring of environmental changes, accurately recording dynamic variations in external conditions [1]. Optical interferometric sensing technology enables the high-precision measurement of various physical quantities by analyzing the relative changes in phase or frequency between two beams of light. This technique has demonstrated high sensitivity to minute fluctuations in the measured signal, thereby validating its effectiveness and reliability in the measurement of physical quantities [2].

For commonly used fiber optic sensors in engineering applications, there has been some theoretical development coupled with further practical validation. Zhu Z et al. have developed an innovative distributed fiber optic sensor for use in landslide monitoring. Utilizing fiber micro-bend loss and optical time–domain reflectometry, the sensor accurately determines the displacement of key points through shear and bending experiments, demonstrating high stability. This offers a reliable and cost-effective solution for landslide monitoring [3]. Buchade P. B. et al. [4,5] investigated the impact of the angle between the emitting and receiving fibers on the performance of dual-fiber intensity-modulated displacement sensors, established a mathematical model based on optical geometry, and discovered that the sensitivity increased 16-fold when the angle was increased from 0° to 30°, revealing the correlation between performance metrics and geometric parameters. Yang C. and Oyadiji S. O. [6] developed a fiber optic displacement sensor based on the Gaussian beam model, the theoretical displacement response was successfully derived, and the consistency between frequency response and displacement vibration pattern was verified. This established the linear prediction range and sensitivity of the sensor in structural displacement monitoring, providing a theoretical and experimental foundation for the study of the relationship between displacement and optical frequency. With the scholars in refs [7,8] utilizing the principle of fiber optic bending loss, the curvature radius of the fiber was increased through a rose-line winding method, thereby enhancing the bending loss. They proposed a theoretical formula to describe the relationship between the circumference of the rose line, the change in fiber optic displacement, and the attenuation of light intensity. Furthermore, they derived a mathematical model for displacement and strain, achieving effective fitting with theoretical and experimental data. Leveraging how the technology of advanced sensors facilitates the convenient measurement of a multitude of required parameters, Liu X. et al. [9] utilized the ABAQUS 6.14 software; a finite element model was constructed, taking into account a comprehensive set of factors including pipelines and geological environments. This multi-dimensional study of pipeline response to ground subsidence provided a theoretical basis and reference values for determining the maximum settlement displacement of pipelines. Additional research has indicated that DFOS exhibit excellent performance in measuring circumferential strain and are highly sensitive to changes in circumferential strain caused by varying degrees of corrosion, that is, alterations in structural dimensions [10]. In the field of settlement monitoring, relevant studies have utilized a combination of indoor and outdoor experiments along with numerical simulation to propose an inversion model for tunnel settlement. Calculations of the settlement at the tunnel crown have been performed. The findings from this research can provide a theoretical and experimental foundation for the application of DFOS technology in the monitoring of tunnel settlement [11]. In the context of tunnel monitoring, there is also the application of strain monitoring technology using distributed fiber optic sensors. This technology has been applied to the deformation monitoring of tunnels, with an analysis conducted on the deformation results of a tunnel case study, demonstrating the feasibility of the technology in such instances [12].

Commonly used strain sensors often have several drawbacks: (1)In the monitoring of caverns, the measurement of hoop strain using strain gauges often results in low efficiency and complex arrangement.(2)Deploying various displacement sensors is challenging in harsh environments.(3)Cavern environments are typically adverse, often facing deep ground and disturbances, leading to poor durability in real-time dynamic displacement monitoring, which cannot operate over extended periods. There is an urgent need for effective advanced measurement and computational methods to address these issues.

Fiber optic sensors demonstrate outstanding performance in numerous applications, offering high precision and reliability. Although the initial investment is relatively high, the long-term maintenance costs of these sensors are lower, with extended service life that effectively reduces failure rates and downtime. In large-scale engineering applications, the real-time monitoring capability of fiber optic sensors enables the timely detection of potential issues, enhancing safety and preventing significant losses. Therefore, from an overall perspective, fiber optic sensors provide excellent cost-effectiveness and are well worth widespread adoption in relevant fields [13].

Therefore, distributed fiber optic sensors, which offer the advantages of accurate, sensitive, and easy measurement, can better meet the monitoring needs of structures in harsh environments. Accordingly, this paper employs the accurate and full-length measurement capabilities of distributed fiber optic sensors to propose a new method for monitoring the displacement of critical points through strain variations. By qualitatively analyzing the reference values through changes in optical properties, a corresponding relationship between key-point displacement and circumferential strain in the cavern model is established, providing a reference for solving the measurement challenges of displacement at certain points in cavern monitoring.

## 2. Theoretical Computational Model

In analyzing the characteristics of sensors, based on the extensive research by many scientists on bending loss, it has been determined that optical power decreases with an increasing transmission distance. In practical applications, the loss of optical fibers is typically expressed in units of “decibels” (dB), which define the attenuation of optical power per unit length of the fiber [14].(1)$$\alpha =\frac{10}{Z}log\frac{P\left(0\right)}{P\left(Z\right)}$$

*α* represents the attenuation coefficient of the optical fiber, with units of decibels (dB) per meter (dB/m), describing the optical power loss per unit length of fiber; *Z* represents the refractive index of the fiber; *P*(0) indicates the optical power at the input end of the fiber; *P*(*Z*) signifies the optical power at the output end of the fiber.

For single-mode fiber transmission, the changes mentioned are accompanied by bending loss and spectral changes. Therefore, a formula for calculating the bending loss of single-mode fibers is employed:(2)$$\alpha_{c}=A_{c}R^{-0.5}exp(-\mathrm{UR})\;A_{c}\approx 30\Delta^{0.25}\lambda^{-0.5}{\left(\lambda_{c}/\lambda \right)}^{1.5}$$

In the formula, Δ represents the difference in refractive index between the core and the cladding; *R* is the radius of curvature of the optical fiber when bent; *λ* denotes the operating wavelength; *λc* signifies the cutoff wavelength [15].

This study analyzed the impact of the bending radius R on the bending loss and spectral characteristics of optical fibers, finding that these characteristics are primarily controlled by changes in the bending radius when the sensor parameters are constant. Based on this, a theoretical model of the relationship between the bending radius and loss and spectral changes was established. By monitoring changes in the fiber optic axis, the study accurately determined the loss and spectral characteristics, then inferred other key parameters. The research proposed the use of fiber optic sensors for structural monitoring, indirectly measuring physical quantities through changes in optical parameters, and established the relationship between bending loss, spectral changes, and displacement through theoretical analysis [16].

The research analysis concluded that the derivation of optical fiber bending loss does not directly employ the coupling model theory. Instead, it involves simplified analyses with some equivalences and assumptions made to calculate the loss characteristics of bent optical fibers, considering that a certain section of the calibrated fiber develops a refractive index change due to bending. Although different scholars present various loss formulas, they fundamentally exhibit similar forms. Therefore, the bending loss calculation formula for single-mode optical fibers (1) and (2) can be expressed in the following form [17]:(3)$$\alpha_{c}=Aexp(-BR)$$

In the formula, *A* and *B* are constants related to the inherent parameters of the optical fiber (such as the core radius, outer radius of the fiber, refractive index difference between the core and cladding, etc.), which can be determined through parameter configuration or calibration experiments.

Existing computational methods are employed to perform equivalent calculations on the curvature of the fiber optic loop, and these methods are applied to the circular arrangement of optical fibers.

By establishing a polar coordinate system with the center of the circle as the origin, the general parametric equations of the circle can be obtained. At this point, the curvature radius of the circle is taken as its own radius. According to the circumference formula for a circle with radius *R*, the coefficient *k* is taken as *2π*. Thus, the relationship between *c* and $\alpha_{c}$ can be derived as follows:

From Equation (3), a relationship can be derived between C and $\alpha_{c}$:(4)$$C=-\frac{k}{B}ln\frac{\alpha_{c}}{A}=-\frac{k}{B}ln\alpha_{c}+\frac{k}{B}lnA$$

At the initial calibration length $L_{0}$ of the fiber optic loop, the corresponding initial calibration length is associated with the initial bending loss, denoted as $C_{0}$ and $\alpha_{c0}$, respectively. When the size of the fiber optic loop decreases, its circumference is also continuously reduced. The deformation of the circular ring can be obtained through coordinated deformation; hence, the displacement of coordinated contraction and the change in bending loss $\Delta \alpha_{c}(\alpha_{c}-\alpha_{c0})$ is as follows:(5)$$\begin{array}{l} S=C_{0}-C_{i}=\frac{k}{B}(ln\alpha_{c}-ln\alpha_{c0})=\frac{k}{B}ln(\frac{\alpha_{c}-\alpha_{c0}+\alpha_{c0}}{\alpha_{c0}})\\ =\frac{k}{B}\left[ln(\alpha_{c}-\alpha_{c0}+\alpha_{c0})-ln\alpha_{c0}\right]=\frac{k}{B}ln(\Delta \alpha_{c}+\alpha_{c0})-\frac{k}{B}ln\alpha_{c0} \end{array}$$

Let the coefficients $\frac{k}{B}=m$ and $\alpha_{c0}=n$ be the constants to be determined; then, the expression for the coordinated contraction displacement S and $\Delta \alpha_{c}$ is given by the following relationship:(6)$$S=mln\left(\Delta \alpha_{c}+n\right)-mln(n)$$

From the previous equation, a relationship between the coordinated contraction of the optical fiber and the change in bending loss is derived. Reasoning the relationship between the coordinated contraction displacement *S* and the vertical displacement, since the optical fiber undergoes coordinated deformation with the circular ring, the deformation value changes within the circumference of the ring. Under uniform pressure, the curve of the circular ring after compression has an area that closely approximates an elliptical area; thus, the deformed curve after compression is approximated as an elliptical curve (using a segmentation method). At this juncture, the regional circumference $L_{1}$ changes to regional circumference $L_{2}$ as depicted in Figure 1.

From this, variable parameters can be established to theoretically deduce the relationship between the coordinated contraction displacement of fiber deformation and vertical displacement:(7)$$\begin{array}{c} L_{1}=2\pi R/4\\ L_{2}=\left[2\pi R+4\left(R-x\right)\right]/4 \end{array}$$(8)$$S=\left|L_{1}-L_{2}\right|$$

Combining the above formulas, we can deduce the following:(9)$$S=\frac{m}{4}ln(\Delta \alpha_{c}+n)-\frac{m}{4}ln\left(n\right)$$

According to Equation (10), the bending modulation mechanism of the fiber optic loop exhibits a logarithmic relationship with displacement loss. Mathematical derivation indicates that there is a correlation between optical characteristics and the displacement measurement point. However, due to the logarithmic relationship, simple parameter configuration cannot precisely calibrate the sensor.

Therefore, when analyzing and designing the fiber optic transmission system in this study, it was necessary to independently consider both bending loss and spectral shift and to take appropriate measures according to specific conditions to reduce loss and maintain the stability of the optical signal’s frequency or wavelength. There was still considerable uncertainty in determining the bending loss; however, a more stable numerical relationship could be obtained through the relationship between spectral change and strain.

This study employed the differential method for the quantitative analysis of strain and displacement, applicable to distributed fiber optic sensors with continuous measurement points. The differential equation method, as a mathematical tool, is capable of describing and analyzing the numerical change patterns of discrete variables. In the context of fiber optic sensors, this approach can accurately capture the displacement changes at the endpoint x when the fiber is bent. By solving the first- and second-order differential equations, the coordinated change values of the fiber optic bending displacement can be obtained, thereby accurately reflecting the displacement changes at specific points [18].(10)$$\Delta f=\frac{f_{x+h}-f_{x}}{h}$$(11)$$\Delta (\Delta f)=\frac{1}{h}(\frac{f_{x+2h}-f_{x+h}}{h}-\frac{f_{x+h}-f_{x}}{h})=\frac{1}{h^{2}}(f_{x+2h}-2f_{x+h}+f_{x})$$

In the formula, *h* represents the interval of the selected measurement points and *f* denotes the displacement of the selected discrete measurement points. The selected measurement points and the definitions and values of the related parameters are shown in Figure 2:

Under the coordinated deformation of the fiber optic and the selected area, the relationship between the obtained strain and the radius of curvature is as shown in Equation (13) [18].(12)$$\frac{\varepsilon (x)}{r}=\frac{1}{\rho (x)}$$

*r* represents the distance from the neutral axis to the boundary and *ρ* denotes the radius of curvature. Similarly, for the selected region, the relationship between the curvature radius *ρ*(*x*) and the deflection *f*(*x*) can be expressed as follows:(13)$$\frac{1}{\rho (x)}=\frac{d^{2}f(x)}{dx^{2}}$$

Combining the above, the relationship between the displacement at the strain measurement point and the circumferential strain can be deduced as follows:(14)$$f(x)=\frac{1}{r}\iint \varepsilon (x)dxdx$$

Under the condition of equal cross-section for the range of calculation selected, the bending stiffness within this annular range remains constant. Therefore, by utilizing the relationship between displacement and bending moment, we find the following:(15)EIf″(x)=M(x)

The experiment utilized a distributed fiber optic sensor for continuous strain monitoring and derived the strain–displacement relationship through discretization analysis. By plotting the data curves of selected measurement points, the relationship between strain and displacement was visually presented. Furthermore, based on a specific formula (Equation (17)), a mathematical relationship between the second-order differential displacement and the strain at the measurement points was established.(16)$$\begin{array}{c} \Delta (\Delta f)=\frac{1}{h^{2}}(f_{x+2h}-2f_{x+h}+f_{x})=\frac{M(x)}{EI}=\frac{\varepsilon_{i}}{r} \end{array}$$

The relationship can be represented by a matrix equation as follows:(17)$$\frac{r}{h^{2}}\left\{\begin{array}{cccccc} 1 & -2 & 1 & 0 & \cdots & 0\\ 0 & 1 & \ddots & \ddots & \ddots & 0\\ \vdots & \ddots & \ddots & \ddots & \ddots & \vdots \\ \vdots & \ddots & \ddots & \ddots & \ddots & \vdots \\ \vdots & \ddots & \ddots & \ddots & \ddots & 0\\ 0 & \cdots & 0 & 1 & -2 & 1 \end{array}\right\}\left\{\begin{array}{c} f_{x}\\ f_{x+h}\\ \vdots \\ f_{x+(n+1)h} \end{array}\right\}=\left\{\begin{array}{c} \varepsilon_{1}\\ \varepsilon_{2}\\ \vdots \\ \varepsilon_{n} \end{array}\right\}$$

Considering the bonding sites of the strain measurement tube as fixed ends and setting the boundaries accordingly, $x=0,\;f_{0}=f_{h}=0$. The relationship between displacement and strain is determined by inverting an invertible matrix within the equation. When selecting measurement points, it is essential to account for coordinated deformation and the extent of deformation to make adjustments within a specific range, ensuring the accuracy of the formula. This approach allows for the precise calculation of the strain and displacement fields within the fiber optic sensor’s measurement domain, taking into account the boundary conditions and the material’s mechanical behavior.(18)$$\begin{array}{c} \left\{\begin{array}{c} f_{h}\\ f_{2h}\\ \vdots \\ f_{nh} \end{array}\right\}=\frac{Ah^{2}}{r}{\left\{\begin{array}{cccccc} 1 & 0 & 0 & \cdots & \cdots & 0\\ -2 & 1 & \cdots & \cdots & \ddots & \vdots \\ \vdots & \ddots & \ddots & \ddots & \ddots & \vdots \\ \vdots & \ddots & \ddots & \ddots & \ddots & \vdots \\ \vdots & \ddots & \ddots & \ddots & \ddots & 0\\ 0 & \cdots & 0 & 1 & -2 & 1 \end{array}\right\}}^{-1}\left\{\begin{array}{c} \varepsilon_{1}\\ \varepsilon_{2}\\ \vdots \\ \varepsilon_{n} \end{array}\right\}+B \end{array}$$

Further deformation analysis involves examining how the relationship evolves as the material undergoes additional strain. This typically includes the following:(19)$$\left\{\begin{array}{c} \varepsilon_{1}\\ \varepsilon_{2}\\ \vdots \\ \varepsilon_{n} \end{array}\right\}=C\left\{\begin{array}{cccccc} 1 & 0 & 0 & \cdots & \cdots & 0\\ -2 & 1 & \cdots & \cdots & \ddots & \vdots \\ \vdots & \ddots & \ddots & \ddots & \ddots & \vdots \\ \vdots & \ddots & \ddots & \ddots & \ddots & \vdots \\ \vdots & \ddots & \ddots & \ddots & \ddots & \vdots \\ 0 & \cdots & 0 & 1 & -2 & 1 \end{array}\right\}\left\{\begin{array}{c} f_{h}\\ f_{2h}\\ \vdots \\ f_{nh} \end{array}\right\}+D\left\{\begin{array}{cccccc} 1 & 0 & 0 & \cdots & \cdots & 0\\ -2 & 1 & \cdots & \cdots & \ddots & \vdots \\ \vdots & \ddots & \ddots & \ddots & \ddots & \vdots \\ \vdots & \ddots & \ddots & \ddots & \ddots & \vdots \\ \vdots & \ddots & \ddots & \ddots & \ddots & \vdots \\ 0 & \cdots & 0 & 1 & -2 & 1 \end{array}\right\}$$

This can be characterized as follows:(20)$$\left\{\begin{array}{c} \varepsilon_{1}\\ \varepsilon_{2}\\ \vdots \\ \varepsilon_{n} \end{array}\right\}=E_{1}\left\{\begin{array}{c} f_{h}\\ f_{2h}\\ \vdots \\ f_{nh} \end{array}\right\}+E_{2}$$

*E*_1_ and *E*_2_ are both *n*-th order coefficient matrices.

Through the aforementioned formulas, it can be deduced that there is a correlation between the displacement at key points and the strain in the local area. When there are a sufficient number of measurement points, the relationship between displacement and circumferential strain is approximately linear. Further empirical validation is conducted to derive the corresponding formulas.

## 3. Experimental Process

### 3.1. Instrument Selection

#### 3.1.1. Measurement System

The experimental measurement instrument utilizes a Luna ODiSI-A sensor model as Figure 3, which employs the scanning wavelength interferometry method to interrogate the fiber optic sensors. The ODiSI-A fiber optic distributed sensing system is capable of achieving full distributed strain and temperature measurements with millimeter-level spatial resolution, with a maximum sensing length of up to 50 m, making it a versatile testing device applicable for various scenarios. This system is based on Optical Frequency Domain Reflectometry (OFDR) technology and utilizes the single-mode fiber as the sensor, providing novel solutions for the testing of advanced materials and complex structures in the 21st century. A schematic of the ODiSI optical network is depicted in Figure 3, where the optical system consists of a tunable laser source (TLS), an interferometer (including the sensor), and a detector. The instrument’s maximum rated output power is 9.0 mW, with the internal laser module’s maximum rated output power at 20.0 mW, emitting wavelengths ranging from 1510 to 1570 nm, and a strain measurement accuracy of ±60 με. This enables comprehensive positional measurements throughout the entire course. In the academic context, the fiber grating pitch is 250 μm, the total length of the fiber is 2 m, the effective length is 1.4 m, and optical matching gel is applied at the termination end.

The sensor’s strain changes cause measurable variations in the scattered light of the optical fiber, which are determined by comparing them with reference values from the optical fiber to ascertain the state of the fiber. To ensure measurement accuracy, the temperature is kept constant during the experiment, negating its short-term impact on the strain data. In strain sensor mode, the Rayleigh scattering characteristics are measured and stored as baseline values. After deformation, the optical fiber experiences spectral shifts, which differ from the baseline, simulating the shift in the resonant wavelength of a Bragg grating [19,20,21].(21)$$\frac{\Delta \lambda }{\lambda }=-\frac{\Delta \nu }{\nu }=K_{T}\Delta T+K_{\varepsilon }\varepsilon$$

In the equation mentioned, λ and ν represent the average optical wavelength and frequency, respectively, while $K_{T}$ and $K_{\varepsilon }$ are the temperature and strain calibration constants, respectively. The default values for these constants are set in the sensor configuration software *OdiSI-disk* and are suitable for most germanium-silicate core fibers, with the values given as $K_{T}=6.45\times 10^{-6}$ and $K_{\varepsilon }=0.780$ for the constants. From the equation, it can be deduced that under constant temperature conditions, the spectral change is linearly related to the strain [22,23].

The ODiSI system is noted for its high spatial resolution. However, it is prone to thermal cross-sensitivity due to temperature variations during strain measurements. To mitigate this, precise calibration and data fusion techniques are employed to isolate temperature and strain effects, ensuring accurate measurements [24,25,26]. The ODiSI system’s automated calibration and versatile measurement modes provide reliable strain and temperature data in stable thermal environments, which is crucial for structural health monitoring and smart structure design. Additionally, in scenarios with significant temperature variations, the application of novel encapsulation techniques or the installation of temperature sensors for subsequent data compensation can enhance the accuracy of the data obtained in the experimental environment [27,28,29].

Temperature and strain both lead to changes in the scattered light of the optical fiber. Through indoor experimental planning, with the room temperature kept constant, the ODiSI can provide strain measurements along the length of the sensor. The strain can be expressed as follows:(22)$$\varepsilon =-\frac{\overset{-}{\lambda }}{cK_{\varepsilon }}\Delta V$$

Here, $\overset{-}{\lambda }$ is the central wavelength of the scan, *c* is the speed of light, $K_{\varepsilon }$ is the strain constant, and ΔV represents the spectral shift. Thus, it can be concluded that there is a linear relationship between strain and spectral offset [30].

By applying Equation (22) and the aforementioned theoretical formulas, it can be deduced that there exists a linear relationship between spectral shift and key-point displacement. The specific coefficients to be determined are related to the sensor’s configuration. By analyzing the relationship between strain and key-point displacement, the correlation between spectral variation and key-point displacement can be obtained. Therefore, the subsequent analysis of the relationship between strain and key-point displacement yields the association [31,32].

#### 3.1.2. Device System

The testing equipment was divided into a loading system and a measurement system. For the loading process, quasi-static uniaxial compression was selected. The loading system employed an HUT160D microcomputer-controlled electrohydraulic servo universal testing machine with a specification of 1000 kN and an accuracy class of 0.5. During the test, a black sandstone specimen with dimensions of 0.3 m in height, 0.3 m in width, and 0.15 m in thickness was subjected to multiple displacement-controlled tests. After the test began, image capture and loading were initiated simultaneously at a loading rate of 0.1 mm/min. Once the loading was completed, the images were obtained for output analysis and the final compression displacement was recorded as 1.01 mm.

The Basler A602f camera is a high-performance industrial camera with a resolution of 656 × 492 pixels, providing a field of view of 656 × 491 pixels. This camera features pixel sizes of either 8 or 12 bits and supports an image frequency of up to 25 frames per second, making it suitable for high-speed imaging applications. Basler provides the Framegrabber SDK and Pylon Viewer software for this camera, facilitating image capture and analysis for users.

Displacement sensors were installed to measure the displacement of key points. In this study, the top point of the cavern was selected as the key point. Subsequently, optical fiber sensors were installed along the central circumference of the cavern. The sensors were bonded with a special adhesive to ensure coordinated deformation between the structure and the fibers. At the end of the fiber extending from the cavern, a fiber optic signal matching paste was used as the termination point. The transmitting end was then inserted into the demodulator to complete the calibration of the fiber optic sensors. The overall layout is shown in Figure 4.

#### 3.1.3. DIC Analysis

In the experiment, we applied densely scattered pattern stickers on the surface of the specimen for quasi-static testing and used Digital Image Correlation (DIC) technology to monitor the strain field changes on the specimen’s surface, comparing them with numerical simulation results for validation. Prior to the experiment, the position and focus of the industrial camera were adjusted to ensure that image quality met the required precision standards. DIC technology enabled us to observe the gradual increase in strain under displacement control, facilitating the visualization of critical areas for analysis. The strain field changes are depicted in Figure 5.

We used the physical field analysis software GOM (Optical 3D metrology and inspection software developed by GOM GmbH (Braunschweig), a ZEISS Group company.) for Digital Image Correlation (2D-DIC) cloud analysis. It can be observed from the image that the strain change was significant in the area where displacement was applied at the upper end, and it corresponded with the numerical simulation.

### 3.2. Experimental Result Analysis

According to the strain results obtained after the test, the circumferential strain increased linearly with time. Subsequently, by analyzing the relationship between the key-point displacements within the region and the circumferential strain, a scatter plot could be obtained within the allowable error range. The linear fitting of this scatter plot yielded the graph shown in Figure 6.

From the fitted graph, it is evident that there was a clear linear relationship between the two variables, and the displacement trend could be deduced from the strain variation. With an $R^{2}$ value of 0.9852, which was close to 1, the fit could be considered to have a high degree of accuracy. The equation of the fitted curve was as follows:(23)$$\varepsilon_{h}=9.539\times 10^{-5}f-4.56\times 10^{-6}$$

## 4. Numerical Simulation Analysis

The test conditions and the data obtained were analyzed. The numerical simulation software LS-DYNA (Finite Element Analysis Software) was used to conduct numerical simulations on the specimens in the tests and the results were compared with the experimental data to obtain a reliable numerical simulation model for further extended experiments. The numerical simulation could predict the strain distribution and displacement field of the structure under load, identify and verify the high-strain-sensitive areas (such as the vicinity of the underground chamber), and thus guide the layout and density optimization of fiber optic sensors to ensure the effective monitoring of key areas. Meanwhile, the simulation could quickly assess the influence of different material parameters (such as shear modulus) and structural dimensions (such as the diameter of the chamber) on the response (see Section 5), helping determine the range of variables that needed to be strictly controlled in the experiments. Finally, the feasibility of the loading scheme (such as the amplitude of displacement control) and the correctness of the theoretical model could be verified in advance to avoid equipment damage or data failure due to unreasonable design during the experimental process.

### 4.1. Finite Element Model Establishment

Utilizing TrueGrid (Mesh Generation Preprocessing Software), a specimen model was established based on the circular opening model, with dimensions determined to ensure a sufficiently large radius for the fiber optic sensor to respond to bending. The designed small-scale experimental model, as shown in Figure 7, consisted of a cubical block with an opening, measuring 0.3 m × 0.3 m × 0.15 m, and a chamber diameter of 0.2 m. The mesh in the areas of particular structural interest was appropriately adjusted. After conducting a mesh sensitivity analysis and verification through computational assessment, a mesh size of 10 mm was selected. The model comprised approximately 40,000 elements.

### 4.2. Material Parameter Configuration

The rock material model employed the MAT_RHT model, using the unit system of mm-ms-kg-GPa. The detailed material parameters were set in reference to the sensitivity and determination method study of the main parameters of the RHT model. This model was capable of matching the structural response required for this experiment. The key parameters are listed in the Table 1 according to the purpose of the experiment.

### 4.3. Numerical Simulation Setup

After analyzing the test environment, the boundary conditions were determined for the simulation loading. The displacement was applied using the BOUNDARY_PRESCRIBED_MOTION keyword, which defines a prescribed motion (displacement, velocity, or acceleration) for nodes or node sets. The BOUNDARY_SPC keyword was used to apply constraints to the selected node surfaces. Friction was applied to the upper and lower surfaces to match the displacement and constraint modes of the test, ensuring strain transfer.

Based on the above settings, numerical simulations were performed to obtain model data under different displacement controls. The post-processing in LS-PrePost (Preprocessing and postprocessing software designed for LS-DYNA) was used to export the node information, element stress, and node displacement of the structure after displacement control. In the explicit calculation, the time step was elongated to ensure stable computation, with the kinetic energy dissipation approaching zero in each equilibrium segment. The numerical simulation under displacement control was conducted for the structure that met the test conditions, following the aforementioned methods.

### 4.4. Analysis of Numerical Simulation Results

A comparison was made between the experimental and numerical simulation results under compression, with the numerical simulation providing a description and analysis of the stress and deformation patterns in areas that were not directly observable in the experiment. Key displacement values obtained from the numerical simulation correlated with circumferential strain, with the relationship established through the z-direction displacement data that changed over time.

Figure 8 presents the deformation characteristics of the numerical simulation corresponding to the experiment. The deformation characteristics calculated by the numerical simulation were consistent with the corresponding experimental phenomena, demonstrating that numerical simulation could reflect the deformation patterns of components under displacement control.

Analysis of the high-speed photography images from the DIC (Digital Image Correlation) technique revealed distinct characteristics of the strain field during the compression phase. It was observable that the strain within the region of interest increased significantly over time. These findings could be compared and analyzed against the later numerical simulation model, and the displacement changes of key points in the experiment could be determined, as shown in Figure 9.

Under the model data representation, the top point was identified as a key point, and it was observed that the displacement at this location had a larger value compared to other circumferential points. By comparing the circumferential strain data with the experimentally measured data, it was found that the range of points with significant strain variation in the key area was relatively consistent. The Local Coord System was used to establish a cylindrical coordinate system to represent the circumferential strain and the displacement values of the key points, thus obtaining the relationship between the two. Since the structure and the applied forces were symmetric, half of the structure’s circumferential strain and key points were taken for the relationship analysis. The simulated scatter plot and fitting curve of the key-point displacement versus circumferential strain were obtained, as shown in Figure 10.

The obtained fitting curve had a slope of 9.679 × 10^−5^, an intercept of −4.678 × 10^−6^, and an R^2^ value of 0.99529. Therefore, the equation of the curve could be expressed as follows:(24)$$\varepsilon_{h}=9.679\times 10^{-5}f-4.678\times 10^{-6}$$

Upon comparing the fitting curves of the two datasets, it was evident that both fell within a linear relationship that was permissible within the bounds of error. This aligned with our theoretical description of the relationship between the two, thereby enabling the acquisition of a credible numerical simulation model that characterized the experiment. The comparative curves of experimental and numerical simulation results are shown in Figure 11.

## 5. Multivariate Analysis

### 5.1. Numerical Relationships of Different Materials

By altering the material parameters of the model to obtain the relationship between key-point displacement and circumferential strain under materials of different properties, and in conjunction with the numerical simulation analysis from Section 3, key parameters were changed, specifically the elastic shear modulus in this case. The study investigated whether the linear relationship obtained was universally applicable to different materials and structures during the elastic phase while also analyzing changes in the relationship coefficients under various conditions. To this end, multiple factors in Table 2 were designed for numerical simulation analysis to explore these variations.

We subjected the numerical simulation models under different parameter factors to the same displacement control as described in the previous section, during the small deformation elastic phase. We acquired the data after parameter variation and performed curve fitting for further comparison.

### 5.2. Numerical Analysis of Materials with Different Shear Moduli

Using different shear moduli as criteria for different materials, the impacts of various shear moduli on the coefficients are discussed. A decrease in the shear modulus implies that under the same external force, a material’s deformability will increase, leading to greater deformation or an increased rate of deformation. The shear modulus is an important parameter that describes the material’s stiffness. When the shear modulus is reduced, the material’s stiffness also decreases accordingly, making the material more prone to deformation. Multiple factor tests were conducted on the material in our study, and several control groups were taken to obtain multiple sets of data for key-point displacement and circumferential strain for fitting. By analyzing the changes in the coefficients set in the theoretical formula, the effects of shear moduli of 10.7 GPa, 12.7 GPa, 16.7 GPa (initial), 18.7 GPa, and 20.7 GPa on the slope and intercept of various models were explored through parameter comparison.

The MAT_RHT model was modified by altering the shear modulus parameter, denoted as “Shear”, to obtain different fitting curves that intuitively revealed the patterns of change. By fitting, the slope and intercept of the curves were obtained. As the shear modulus increased, the slope and intercept were analyzed to further confirm the pattern of change in the coefficients that were to be determined.

When the shear modulus changed, the corresponding circumferential strain and key-point displacement were obtained through the numerical simulation model. The relationship between the two was analyzed, and a curve was fitted to obtain the fitting curves for different shear moduli for comparison.

Assigning different parameter numbers facilitated the comparison of images and data, as listed in Table 3.

The obtained fitting curve is shown in Figure 12.

By examining Figure 12 and the magnified local data of the curve, it can be concluded that as the shear modulus increased, the slope of the fitting curve notably decreased, and the absolute value of the intercept diminished. In summary, the slope and intercept of the fitting straight line were clearly related to the shear modulus; hence, the coefficients within the straight line were also correlated with it.

### 5.3. Numerical Analysis of Materials with Different Hole Diameters

After altering the diameter of the holes in the model and applying the same displacement control to the same material, the relevant fitting curves were obtained as shown in Figure 13.

It was observed that when there was a significant change in the hole diameter, that is, after the boundary effect changed, the circumferential strain still exhibited a pronounced and reliable linear relationship with the key-point displacement. Therefore, the variation in the slope and intercept of the curves for various hole diameter sizes was obtained, suggesting that the size of the hole diameter had a direct relationship with the relevant parameters of the curve. The parameter variations are shown in Table 4.

Therefore, by expanding the experimental and numerical simulation parameters, a significant linear relationship between displacement and hoop strain was obtained. The deformation field of the specimen was captured using DIC technology to correspondingly validate the numerical simulation, further obtaining the changes in displacement and hoop strain produced under different conditions. Consequently, hoop strain was utilized to predict and monitor the critical point displacement within the chamber, providing a more convenient method for studying the trend of displacement changes.

### 5.4. Numerical Analysis of Structures Under Large Deformation

After conducting the analysis under the conditions of small deformation in the elastic phase and obtaining theoretically consistent conclusions, numerical simulation analysis was performed for the plastic phase, as well as for large deformation and even failure states. The structure was loaded under displacement control, allowing it to enter the non-elastic phase until failure, and the relevant data curves were obtained, as shown in Figure 14.

Through the analysis of the curve, it was observed that after passing through the elastic region, the strain state underwent significant changes, transitioning into the plastic phase after a period of loading until the failure stage, where a substantial decrease in strain occurred. At this point, other corresponding curves with different slopes and outcomes were required to characterize the situation, as the stable linear correlation of the elastic phase could not be sustained, and the relationship began to tend towards nonlinearity, not meeting the consistent linear relationship that was previously stable.

Meanwhile, as a laboratory-based experiment conducted under controlled conditions rather than harsh environments, the conclusions of this study are applicable to moderate temperature and humidity ranges. The nonlinear effects under extreme conditions require further optimization through additional extreme-condition experiments, simulations, or reinforcement-learning-based adaptive algorithms.

## 6. Conclusions **and Outlook**

(1).This study analyzed the behavior of rock specimens with holes under uniaxial compression, utilizing fiber optic sensors for the precise measurement of key-point displacement and circumferential strain. By correlating optical parameters with displacement through theoretical formulas and analyzing models with different shear moduli and hole structure characteristics through numerical simulation, the research explored the impact of material and structural properties on the relationship between key-point displacement and spectral changes, achieving a novel monitoring method for caverns.(2).Based on the theoretical analysis of bending loss, a theoretical model correlating the displacement of key points with circumferential strain was derived. This theoretical calculation can help expand the performance of fiber optic sensors in practical applications and provides a theoretical basis for their further development and application. Further analysis revealed that the undetermined parameters in the theoretical model were correlated with material properties or structural characteristics.(3).Digital Image Correlation (DIC) technology revealed the variation patterns of the strain field in the specimen. By comparing the experimental results with numerical simulation results, a significant linear relationship and high fitting degree between the displacement of key points and the circumferential strain were confirmed. This finding validated the effectiveness of the theoretical analysis and provided a reliable numerical simulation model basis for analyzing critical deformation points and data processing.(4).In response to changes in material properties, the numerical simulation results indicated that as the shear modulus increased, the slope and absolute value of the intercept of the fitting curve for key-point displacement and circumferential strain decreased significantly. However, the fitting curve between key-point displacement and circumferential strain still exhibited a significant linear relationship. The slope ranged from 9.81 × 10^−5^ to 9.60 × 10^−5^, and the intercept changed from −4.70 × 10^−6^ to −4.61 × 10^−6^, demonstrating that the parameters of the fitting curve were correlated with the material modulus.(5).In response to changes in the hole diameter, the numerical simulation results indicate that as the hole diameter increased, the slope and intercept of the fitting curve for key-point displacement and circumferential strain increased significantly. The slope ranged from −4.697 × 10^−4^ to −9.679 × 10^−5^, and the intercept changed from −9.058 × 10^−6^ to −4.678 × 10^−6^. The fitting curve between key-point displacement and circumferential strain still exhibited a significant linear relationship, indicating that the parameters of the fitting curve had a strong correlation with structural characteristics (hole diameter).

This study investigated the deformation monitoring mechanism of cavern-containing rock masses under uniaxial compression through integrated fiber optic sensing and multiscale numerical simulations. A displacement–strain optical response model was developed based on bending loss theory, with experimental verification confirming a high-precision linear correlation between key-point displacement and circumferential strain during the elastic phase (slope range: 9.60 × 10^−5^ to 9.81 × 10^−5^; goodness-of-fit R^2^ > 0.98). Numerical simulations revealed that each 1 GPa increase in material stiffness (shear modulus) reduced the absolute slope value by 2.1% while cavity diameter expansion induced a more pronounced reduction in slope magnitude, demonstrating coupled material-structural-optical parametric interactions. The proposed hybrid monitoring methodology, combining Digital Image Correlation (DIC) validation with fiber optic measurements, achieves submillimeter displacement resolution (0.01 mm precision), establishing a theoretically rigorous framework for the stability assessment of underground caverns. This breakthrough provides an innovative solution for health diagnostics and disaster early-warning systems in geotechnical engineering, effectively addressing the limitations of conventional monitoring techniques.

Research on the long-term monitoring performance of fiber optic sensors serves as a critical supplement to this work. We plan to conduct continuous monitoring experiments over 1–3 months in follow-up studies, systematically analyzing hardware stability, signal continuity, and environmental impacts to comprehensively evaluate the performance of this fiber optic sensing method during prolonged operation. This will provide more robust scientific evidence for reliable engineering applications of fiber optic sensing technology while elucidating performance evolution patterns and potential limitations under extended use.

## Figures and Tables

**Figure 1 sensors-25-02619-f001:**
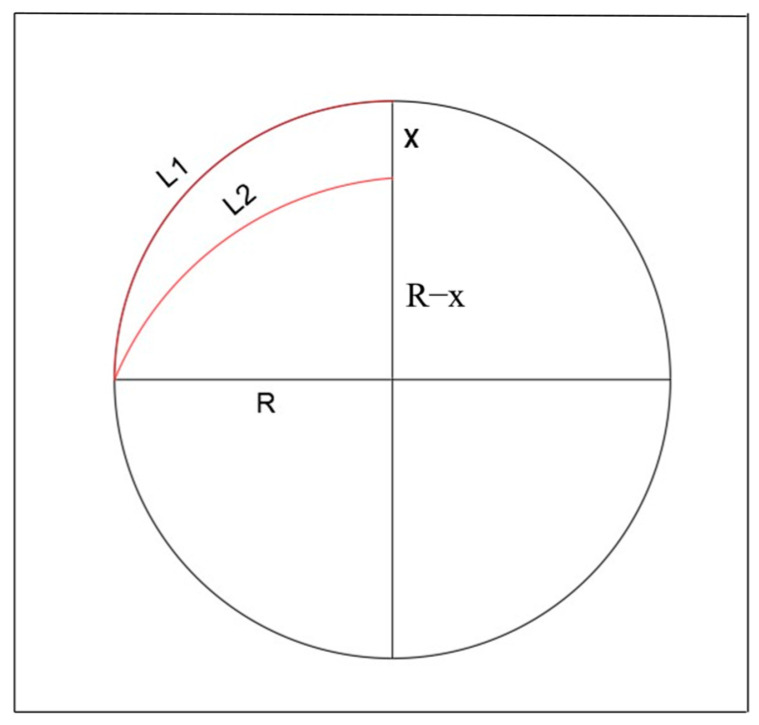
Schematic diagram for calculating the approximate perimeter of the theoretical model region.

**Figure 2 sensors-25-02619-f002:**
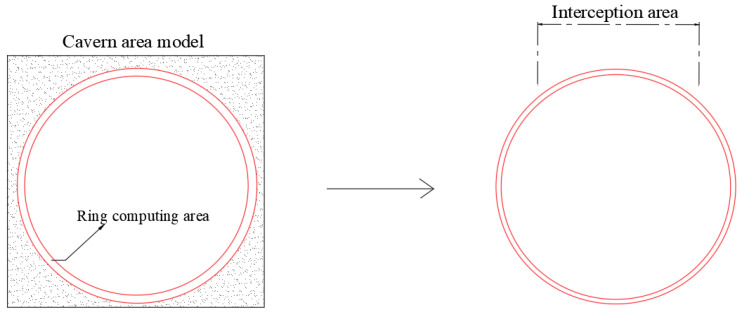
Schematic diagram of displacement and strain calculation in the strain-sensitive region.

**Figure 3 sensors-25-02619-f003:**
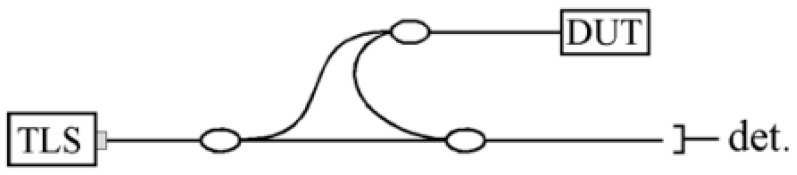
Schematic diagram of the optical network for the OdiSI device.

**Figure 4 sensors-25-02619-f004:**
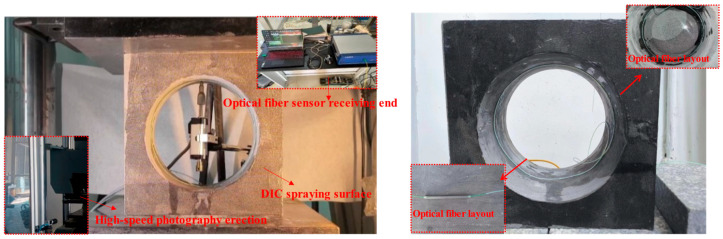
Loading test machine and Luna OdiSI-A sensor and fiber arrangement.

**Figure 5 sensors-25-02619-f005:**
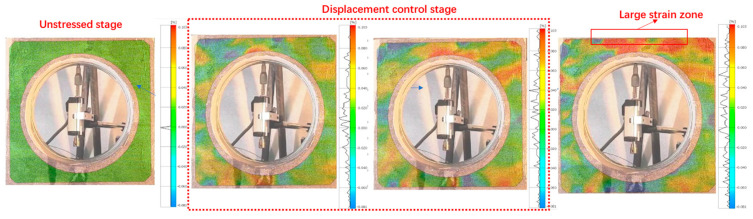
Strain field variation of the DIC coating on the front surface of the specimen.

**Figure 6 sensors-25-02619-f006:**
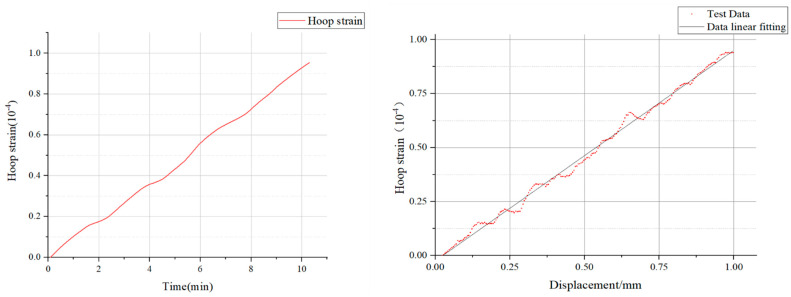
Strain results from fiber optic sensors and the fitting line of experimental data.

**Figure 7 sensors-25-02619-f007:**
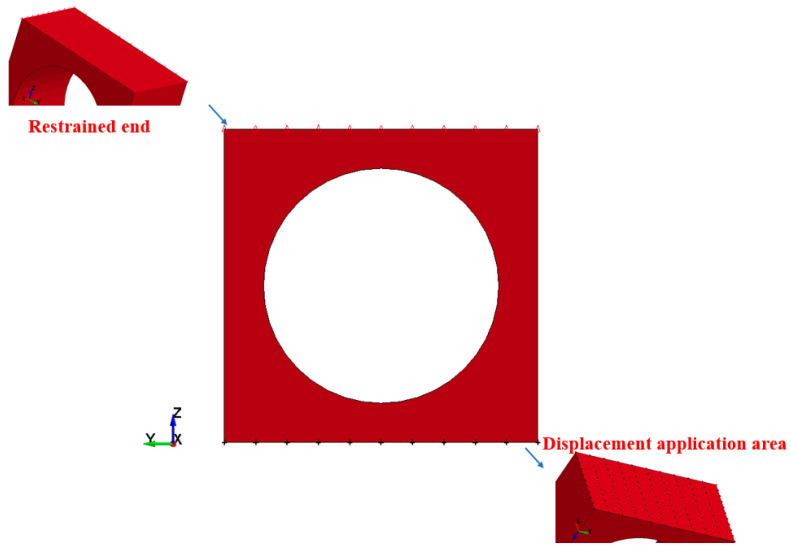
Schematic diagram of applying nodal commands in the numerical simulation model.

**Figure 8 sensors-25-02619-f008:**
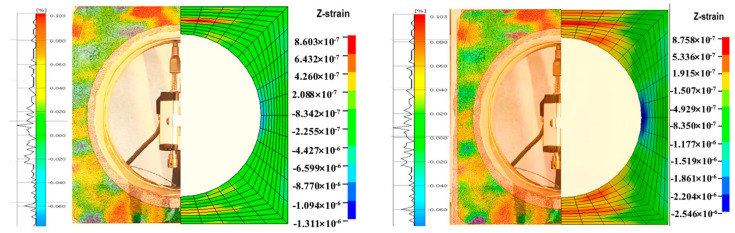
Analysis of corresponding images of DIC and numerical simulation strain contour maps.

**Figure 9 sensors-25-02619-f009:**
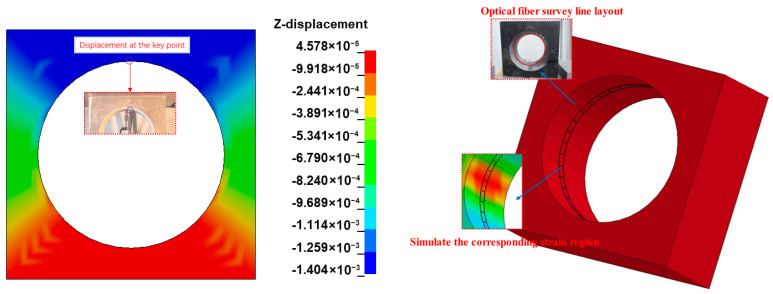
Experimental and numerical simulation of optical fiber layout area.

**Figure 10 sensors-25-02619-f010:**
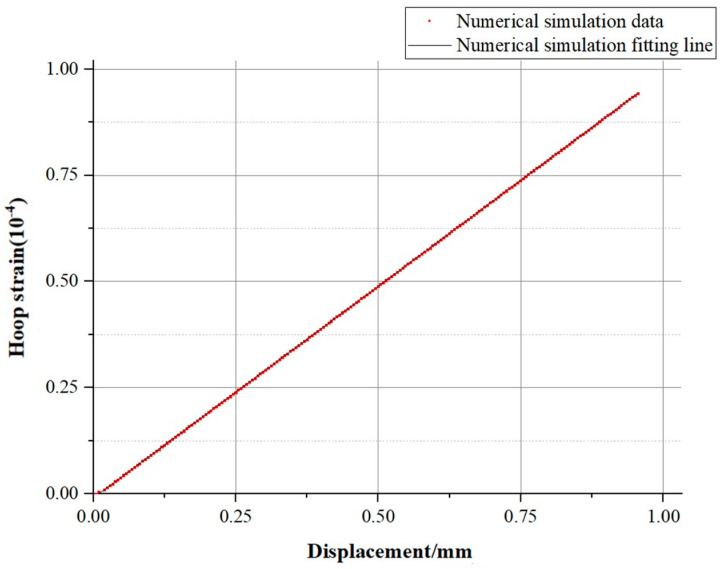
Fitting lines of displacement–strain results from numerical simulation.

**Figure 11 sensors-25-02619-f011:**
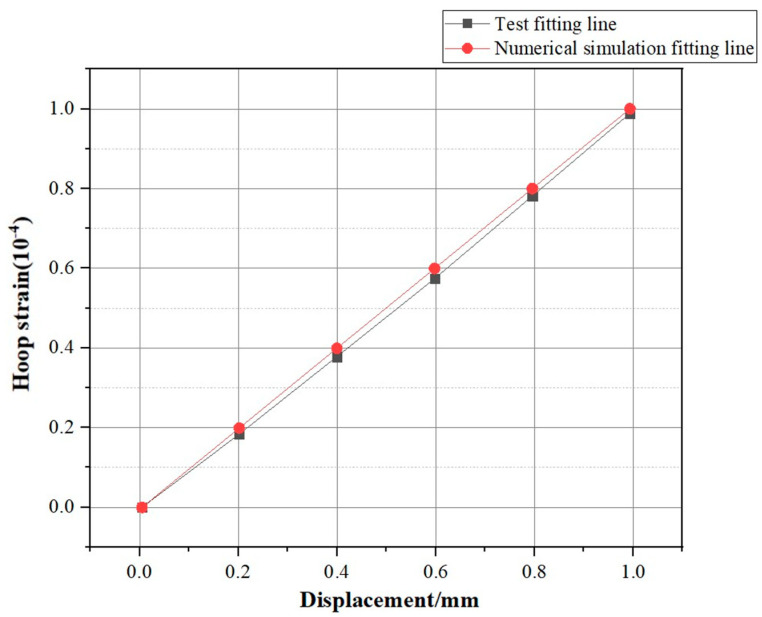
Comparison of experimental and numerical simulation curves.

**Figure 12 sensors-25-02619-f012:**
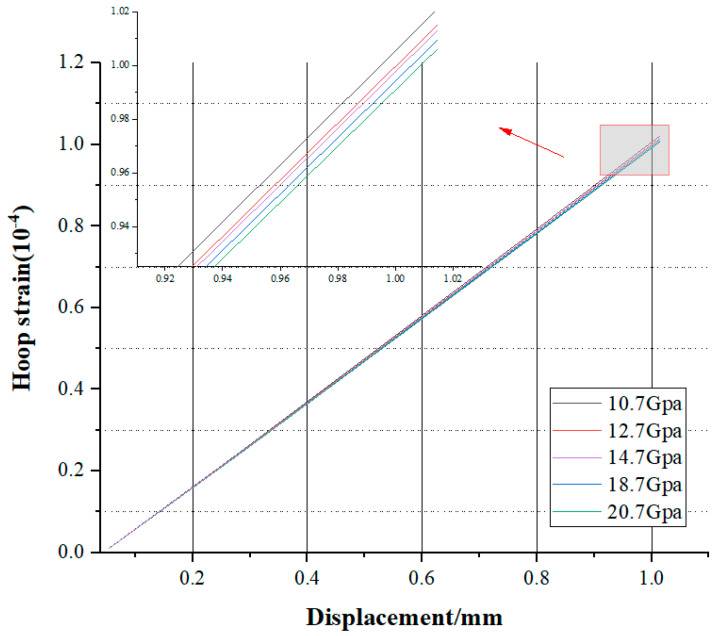
Fitting lines of displacement–strain from numerical simulations with different modulus parameters.

**Figure 13 sensors-25-02619-f013:**
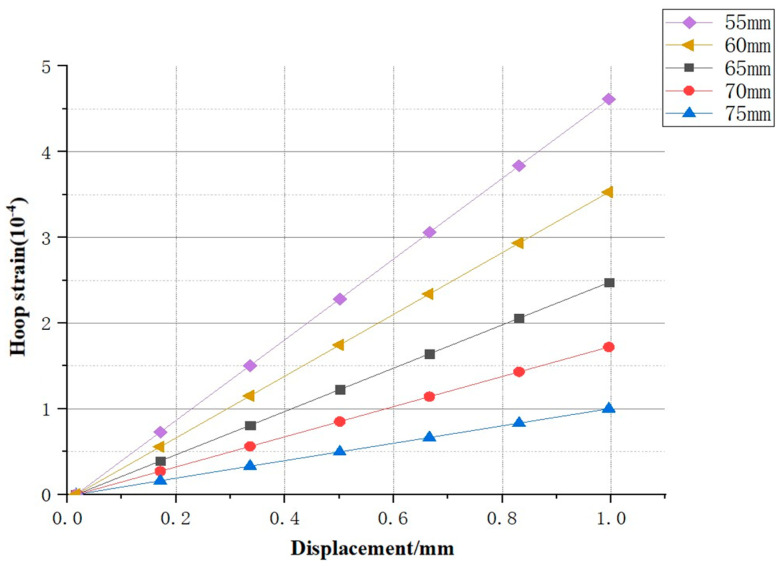
Fitting lines of displacement–strain corresponding to different cavern diameters.

**Figure 14 sensors-25-02619-f014:**
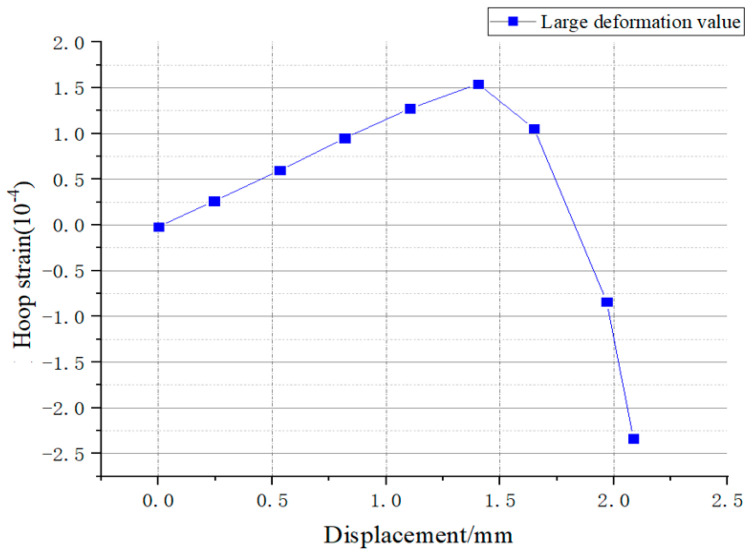
Corresponding relationship data diagram under large deformation control.

**Table 1 sensors-25-02619-t001:** MAT_RHT material key parameters [33].

Density (kg/mm^3^)	Shear Modulus (Gpa)	Uniaxial Compressive Strength (Gpa)
$2.314\times 10^{-6}$	16.7	0.035

**Table 2 sensors-25-02619-t002:** Multi-factor numerical simulation parameter changes.

Number	Diameter of the Hole (mm)	Shear Modulus (Gpa)
N-1 (Initial)	75	16.7
N-2	75	10.7
N-3	75	12.7
N-4	75	14.7
N-5	75	18.7
N-6	75	20.7
N-7	70	16.7
N-8	65	16.7
N-9	60	16.7
N-10	55	16.7

**Table 3 sensors-25-02619-t003:** Parameters of different shear modulus fitting curves.

Number	Shear Modulus (Gpa)	Slope	Intercept	R^2^
N-1 (Initial)	16.7	$9.679\times 10^{-5}$	−4.678 × 10^−6^	0.997
N-2	10.7	$9.810\times 10^{-5}$	−4.709 × 10^−6^	0.989
N-3	12.7	$9.765\times 10^{-5}$	−4.684 × 10^−6^	0.996
N-4	14.7	$9.690\times 10^{-5}$	−4.655 × 10^−6^	0.994
N-5	18.7	$9.640\times 10^{-5}$	−4.630 × 10^−6^	0.988
N-6	20.7	$9.608\times 10^{-5}$	−4.615 × 10^−6^	0.995

**Table 4 sensors-25-02619-t004:** Fitting curve parameters under different hole diameters.

Number	Hole Diameters (mm)	Slope	Intercept	R^2^
N-1 (Initial)	75	−9.67 × 10^−5^	−4.67 × 10^−6^	0.997
N-7	70	−1.62 × 10^−4^	−6.44 × 10^−6^	0.999
N-8	65	−2.43 × 10^−4^	−7.21 × 10^−6^	1
N-9	60	−3.51 × 10^−4^	−8.15 × 10^−6^	0.998
N-10	55	−4.69 × 10^−4^	−9.05 × 10^−6^	0.999

## Data Availability

The data used to support the findings of this study are available from the corresponding author on request.

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
