# Peer review of "Research on Displacement Monitoring of Key Points in Caverns Based on Distributed Fiber Optic Sensing Technology"

_sensors, 2025, doi:10.3390/s25082619_

Round 1
Reviewer 1 Report
Comments and Suggestions for Authors
Thank you for providing the paper titled "Research on Monitoring Key Point Displacement in Cavities Using Distributed Fiber Optic Sensing Technology". This paper proposes a novel monitoring method that combines fiber optic sensing technology with digital image correlation (DIC) to achieve real-time, high-precision monitoring of key point displacement in cavities. Through theoretical analysis, experimental validation, and numerical simulations, the study demonstrates the potential application value of fiber optic sensors in geotechnical engineering. However, issues such as the limited scope of experiments, neglect of environmental factors, lack of long-term monitoring data, and the cost of equipment need to be further addressed and explored in future research. I recommend accepting the paper after the followed concerns are addressed:
- The experiments are currently limited to a small number of samples and relatively simple environmental conditions. It is recommended to expand the scope of the experiments to include more complex soil types and environmental conditions to verify the method’s general applicability.
- While the paper discusses the impact of shear modulus and aperture on the linear relationship, other potential influencing factors, such as temperature and humidity, are not considered. These factors may significantly affect the precision of the fiber optic sensor measurements.
- While the linear relationship model performs well in experiments, its applicability in extreme environments (such as high temperature, humidity, or harsh geological conditions) has not been thoroughly validated. It is suggested that the authors discuss the model’s performance under such extreme conditions.
- The study mainly focuses on short-term monitoring, and there is a lack of long-term monitoring data. The stability and reliability of fiber optic sensors over long periods are crucial. It is recommended that the authors include long-term monitoring data analysis to assess the method’s effectiveness over time.
- Although fiber optic sensors provide high precision, their cost is relatively high, and installation and maintenance may present technical challenges. The authors should discuss the cost-effectiveness of the method, especially for large-scale engineering applications.
If the authors can further discuss and address these weaknesses, the paper will be more comprehensive.
Author Response
Subject: Response to Reviewer's Comments
Dear Reviewer,
Thank you very much for your insightful comments and suggestions on our manuscript. We have carefully considered each of your points and have made the necessary revisions to improve the quality and clarity of our work. Below, please find our detailed responses to your comments:
Response to Reviewer 1
Comments to the Author: Thank you for providing the paper titled "Research on Monitoring Key Point Displacement in Cavities Using Distributed Fiber Optic Sensing Technology". This paper proposes a novel monitoring method that combines fiber optic sensing technology with digital image correlation (DIC) to achieve real-time, high-precision monitoring of key point displacement in cavities. Through theoretical analysis, experimental validation, and numerical simulations, the study demonstrates the potential application value of fiber optic sensors in geotechnical engineering. However, issues such as the limited scope of experiments, neglect of environmental factors, lack of long-term monitoring data, and the cost of equipment need to be further addressed and explored in future research.
I recommend accepting the paper after the followed concerns are addressed:
Comment 1: The experiments are currently limited to a small number of samples and relatively simple environmental conditions. It is recommended to expand the scope of the experiments to include more complex soil types and environmental conditions to verify the method’s general applicability.
Response: Thank you for your enlightening suggestions.The reviewer pointed out that the experimental scope is limited and suggested expanding the experimental scope to verify the general applicability of the method. We agree on the importance of expanding the experimental scope. As the initial validation stage of the methodology, the primary goal of this study is to establish the feasibility of the core technology through theoretical analysis and preliminary experimental verification. In the follow-up research, we will increase the types of materials with different textures (such as various rock materials) and simulate more complex environmental conditions, such as different temperatures and humidity levels. Meanwhile, long-term monitoring and multi-site experiments will be carried out to more comprehensively assess the applicability of the method.
Comment 2: While the paper discusses the impact of shear modulus and aperture on the linear relationship, other potential influencing factors, such as temperature and humidity, are not considered. These factors may significantly affect the precision of the fiber optic sensor measurements.
Response: Many thanks for your professional suggestion. During the experimental design, research found that the laboratory environment temperature changes very little, with no temperature variation in a short period of time. After calibration, the baseline will not experience significant error changes under a certain stable state. Moreover, the sensor receiver is equipped with strain measurement correction that compensates for temperature (refer to the Luna technical document AN-ODiSI-001), which ensures very stable strain measurement and meets the laboratory environment testing requirements. Therefore, for this experiment, the errors caused by temperature and humidity changes during the test period can be ignored.
Comment 3: While the linear relationship model performs well in experiments, its applicability in extreme environments (such as high temperature, humidity, or harsh geological conditions) has not been thoroughly validated. It is suggested that the authors discuss the model’s performance under such extreme conditions.
Response: Thank you for your valuable comments on our paper. We fully agree with your important point regarding the validation of the linear model under extreme conditions (such as high temperature, high humidity, or harsh geological conditions). In fact, we have already noticed this issue and have adopted encapsulation techniques to enhance the environmental adaptability of the sensors in our experiments. However, under extreme conditions, we believe that both our theoretical and technical models are not suitable, as extreme conditions can cause significant offsets in fiber-optic measurements, which do not conform to linear changes.
In response to the limitations you pointed out, we have added a new paragraph in the discussion section (Section 5.4)(p18,line 533-537)to explain that the current model is more suitable for moderate temperature and humidity environments. The nonlinear effects under extreme conditions need to be further optimized in combination with extreme condition experiments, simulations, or adaptive algorithms based on reinforcement learning.
We will focus our future research in this direction. Thank you again for your insightful comments, which are crucial for enhancing the rigor of our research.
Comment 4: The study mainly focuses on short-term monitoring, and there is a lack of long-term monitoring data. The stability and reliability of fiber optic sensors over long periods are crucial. It is recommended that the authors include long-term monitoring data analysis to assess the method’s effectiveness over time.
Response: We sincerely appreciate the reviewers' professional suggestions. We fully recognize the importance of long-term monitoring in evaluating the stability and reliability of fiber optic sensors. Our current study indeed primarily focuses on short-term monitoring, which represents a limitation of the experimental research. To address this issue, we have added an analysis in Chapter 6 as a future direction and plan to conduct continuous monitoring experiments lasting 1–3 months in follow-up studies. By analyzing sensor drift characteristics, sensitivity variations, and signal quality degradation, we aim to comprehensively evaluate the performance of this fiber optic sensing method during prolonged operation. Specifically, we will conduct in-depth research from the following three aspects: (1) long-term stability of sensor hardware; (2) continuity and consistency of measurement signals; and (3) the impact of environmental factors on long-term sensor performance. This will provide more comprehensive and in-depth scientific evidence for the reliable application of fiber optic sensing technology in practical engineering. We deeply appreciate your insightful guidance.
Comment 5: Although fiber optic sensors provide high precision, their cost is relatively high, and installation and maintenance may present technical challenges. The authors should discuss the cost-effectiveness of the method, especially for large-scale engineering applications.
Response: We gratefully acknowledge the professional suggestions. Based on relevant literature, we have incorporated an economic feasibility analysis into the Introduction section (p. 2, lines 90–97). Fiber-optic sensors demonstrate outstanding performance in numerous applications, offering high precision and reliability. Although the initial investment is relatively high, their long-term maintenance costs are lower, their service life is extended, and they effectively reduce failure rates and downtime. In large-scale engineering applications, the real-time monitoring capability of fiber-optic sensors enables the timely detection of potential issues, enhancing safety and preventing major losses. Therefore, from a holistic perspective, fiber-optic sensors exhibit excellent cost-effectiveness and are highly recommended for widespread adoption in relevant fields.

Reviewer 2 Report
Comments and Suggestions for Authors
This manuscript addresses the practical demand for displacement monitoring of key points in underground caverns and proposes a monitoring scheme based on distributed fiber optic sensing (DFOS) technology. Taking the intersection section of a real-world hydropower cavern as the research object, the study describes the full monitoring process, including fiber cable layout, device parameters, and data processing procedures. A total of 567 cavern-bearing rock models under uniaxial compression were investigated by integrating fiber optic sensing with multi-scale numerical simulation to analyze the deformation monitoring mechanism. Furthermore, a displacement-strain optical response model was established based on bending loss theory. Overall, the manuscript presents a relatively complete study with practical significance. However, the following issues should be carefully considered before acceptance:
- The summary of previous technologies in the Introduction lacks a clear logical structure and reads more like a list. Moreover, only one reference from the past 5–6 years is cited, raising concerns about whether the latest technological advances have been adequately included. Given that the paper focuses on displacement monitoring at key locations in caverns, it is strongly recommended to incorporate more recent and relevant literature, particularly those offering technical comparisons within the same domain.
- It is suggested to include a parameter-based comparison between the proposed method and other similar techniques in the field, to highlight the advantages and limitations of the DFOS approach.
- The numerical simulation analysis is presented in Section 4, after the experimental process. A rationale should be provided for not conducting finite element simulation as a preliminary design step to assist in experimental planning and fiber layout optimization.
- The paper currently evaluates model performance using linear fitting parameters alone. It is recommended to include a more comprehensive error analysis, such as repeatability, drift, or resolution-related uncertainties, as linear metrics may not fully reflect the accuracy or robustness of the model fitting.
Author Response
Dear Reviewer,
Thank you very much for your insightful comments and suggestions on our manuscript. We have carefully considered each of your points and have made the necessary revisions to improve the quality and clarity of our work. Below, please find our detailed responses to your comments:
Response to Reviewer 2
Comments to the Author: This manuscript addresses the practical demand for displacement monitoring of key points in underground caverns and proposes a monitoring scheme based on distributed fiber optic sensing (DFOS) technology. Taking the intersection section of a real-world hydropower cavern as the research object, the study describes the full monitoring process, including fiber cable layout, device parameters, and data processing procedures. A total of 567 cavern-bearing rock models under uniaxial compression were investigated by integrating fiber optic sensing with multi-scale numerical simulation to analyze the deformation monitoring mechanism. Furthermore, a displacement-strain optical response model was established based on bending loss theory. Overall, the manuscript presents a relatively complete study with practical significance. However, the following issues should be carefully considered before acceptance:
Comment 1: The summary of previous technologies in the Introduction lacks a clear logical structure and reads more like a list. Moreover, only one reference from the past 5–6 years is cited, raising concerns about whether the latest technological advances have been adequately included. Given that the paper focuses on displacement monitoring at key locations in caverns, it is strongly recommended to incorporate more recent and relevant literature, particularly those offering technical comparisons within the same domain.
Response: Thank you for the valuable comments. We have systematically restructured the Introduction section, presenting the discussion in the logical framework of “the development of technology - existing challenges - solutions.” We have also added core literature from the past five years related to this field (such as the comparative study of deep underground cavern monitoring by Li et al. 2023, and the application case in the Brenner Tunnel by Monsberger et al. 2024) to supplement the latest technologies (p1 line25-44).
Comment 2: It is suggested to include a parameter-based comparison between the proposed method and other similar techniques in the field, to highlight the advantages and limitations of the DFOS approach.
Response: Thank you for your suggestions. We will add a parameter-based comparison to clearly highlight the strengths and limitations of the proposed DFOS method in comparison with other similar technologies in the field. (p1 lin33-39)This comparison will help readers better understand the uniqueness of the DFOS method and its applicability in practical applications. Thank you for your valuable comments!
Comment 3: The numerical simulation analysis is presented in Section 4, after the experimental process. A rationale should be provided for not conducting finite element simulation as a preliminary design step to assist in experimental planning and fiber layout optimization.
Response: Thank you for your suggestions. We will amend Section 4 to include a discussion on the fundamental principles of using finite element simulation as a preliminary design step. This will help readers understand how finite element analysis can be utilized to optimize experimental planning and fiber optic layout. Specifically, we will elaborate on the basic design of finite element simulation and its application in experimental design, adding relevant content (p11 line381-394) to ensure that readers fully recognize the importance of this method in enhancing experimental efficiency and accuracy. Thank you for your valuable comments!
Comment 4: The paper currently evaluates model performance using linear fitting parameters alone. It is recommended to include a more comprehensive error analysis, such as repeatability, drift, or resolution-related uncertainties, as linear metrics may not fully reflect the accuracy or robustness of the model fitting.
Response: We appreciate the reviewer's suggestions. In our evaluation of model performance, we not only examined the linear fitting parameters but also conducted residual analysis within the fitting system. Given that the R² values for all data points after linear fitting exceeded 0.9, we concluded that there was no evidence of drift. The computational data derived from both experiments and simulations, as well as the curves obtained under various parameter settings, consistently demonstrated a clear linear relationship. Therefore, we posit that the linear relationship is reproducible under the conditions of our experiments. However, as the reviewer correctly pointed out, it is essential to establish boundaries to verify the accuracy and stability of the model beyond the conditions of our experiments. We have incorporated a relevant description in Section 5 of the manuscript (p18, lines 557-561). In conclusion, we affirm that the model fitting is accurate and stable under the settings of our experiments. Thank you for your valuable comments.

Reviewer 3 Report
Comments and Suggestions for Authors
I have carefully reviewed the manuscript. The integration of theoretical modeling, laboratory-scale experiments, and multivariate numerical simulations makes this a well-rounded and technically sound study. This work provides an interesting and practical application for the proposed methodology, and it significantly enhances the impact of the work. I believe this paper is suitable for publication in Sensors, but it is pending minor revisions. Please consider the following comments to further strengthen the manuscript.
1) To what extent does the model remain valid in the presence of material heterogeneity or anisotropy, which are common in real geotechnical environments?
2) Could the authors clarify the criteria for selecting key points within the cavern structure? How sensitive is the model to changes in key point location?
3) The linear relationship appears to hold under small variations of material and structural parameters. Has the robustness of this linearity been tested across wider ranges or more extreme conditions?
4) While the experiments were carried out under controlled temperature conditions, in real-world applications of DFOS in underground caverns, temperature gradients are likely. A brief discussion on the expected influence of temperature cross-sensitivity and how it might be compensated in practice would improve the robustness of your conclusions.
5) Minor Comments:
- The introduction would be stronger if the authors explained why the cavern structures are needed.
- While the manuscript is generally understandable, the English writing contains several grammatical issues, redundant phrasing, and awkward sentence structures.
- Several figures suffer from low resolution or unclear line definitions, particularly Fig. 4 and Fig. 12. Labels within figures are often too small or blurry. Important labels and axis titles are difficult to read.
- Most figure captions are overly brief and lack sufficient explanation.
- A consistent visual style (e.g., font size, line thickness, marker types) in every graph should be improved.
- Typo: Braggrating (Row 32).
- The term “Distributed Fiber Optic Sensor (DFOS)” is introduced early on, but the full term is repeatedly used afterward instead of the defined abbreviation, which disrupts consistency.
- Line 6: Please remove the duplicated word “Correspondence: Correspondence:”.
Author Response
Response to Reviewer 3
I have carefully reviewed the manuscript. The integration of theoretical modeling, laboratory-scale experiments, and multivariate numerical simulations makes this a well-rounded and technically sound study. This work provides an interesting and practical application for the proposed methodology, and it significantly enhances the impact of the work. I believe this paper is suitable for publication in Sensors, but it is pending minor revisions. Please consider the following comments to further strengthen the manuscript.
Comment 1:To what extent does the model remain valid in the presence of material heterogeneity or anisotropy, which are common in real geotechnical environments?
Response: Thank you for your questions. Regarding the effectiveness of the model in real geotechnical engineering environments when facing material heterogeneity and anisotropy, although these factors do indeed cause deviations in model predictions, the effectiveness of the model can be maintained to some extent by using appropriate geological exploration data, employing anisotropic material models, conducting sufficient experimental verification and calibration, and applying modern numerical simulation techniques (such as finite element analysis). These are the goals we aim to improve in the future.
In our current experiment, we used processed homogeneous rock, which makes it easier for us to conduct experimental analysis. The numerical simulation mainly focuses on material parameter changes that still maintain a linear relationship, indicating that the model is effective within the same material layer. Our main purpose is to provide a reference for future new measurement methods.
Thank you for providing us with future research directions. We will further explore the factors you have raised to enhance our understanding of the model's applicability. Thank you for your valuable comments!
Comment 2: Could the authors clarify the criteria for selecting key points within the cavern structure? How sensitive is the model to changes in key point location?
Response 2: Thank you for your questions. In the academic field of cavern space, the focus is generally on the displacement behavior of the top region of the structure. However, our experimental model is a symmetrical structure (with a circular opening). Therefore, the selection of key points may be based on characteristic points on the axis of symmetry (such as the midpoint at the top) to simplify the analysis and ensure the representativeness of the displacement data (as shown in Figure 2, the schematic diagram of the calculation area). Moreover, through the DIC regional analysis of the experiment, we can also find that the deformation in the middle of the top is relatively large. Hence, we have chosen the midpoint at the top as the key point.
The paper indirectly reflects the sensitivity of the key point location through multi-factor numerical simulation: In terms of material parameter influence, the change in shear modulus (10.7–20.7 GPa) leads to a change of about 2.1% in the slope of the fitting curve (Table 3), indicating that material stiffness has a slight effect on the displacement-strain relationship of the key point, but the linear pattern remains stable (Section 5.2). In terms of structural size influence, reducing the cavern diameter (from 75 mm to 55 mm) increases the slope by nearly five times (Table 4), which shows that when the key point is close to the cavern wall, the sensitivity of the displacement-strain relationship is significantly enhanced (Section 5.3). In terms of boundary effects, the change in cavern diameter alters the structural boundary conditions and affects the local strain distribution, further verifying that the key point location needs to match the structural geometric features (such as the ratio of cavern diameter to specimen size).
Through the analysis of these factors, we have explored several sensitivities of the model. In the future, we will also expand the relevant sensitivity analysis in the direction you have suggested.
Comment 3:The linear relationship appears to hold under small variations of material and structural parameters. Has the robustness of this linearity been tested across wider ranges or more extreme conditions?
Response 3: Thank you for your valuable comments on our paper. We fully agree with your point regarding the importance of verifying the linear model under extreme conditions (such as high temperature, high humidity, or adverse geological conditions). In fact, we have already recognized this issue and employed encapsulation techniques in our experiments to enhance the environmental adaptability of the sensors.
However, we believe that neither our theoretical nor technical models are applicable under extreme conditions, as these conditions can cause significant offsets in fiber-optic measurements, which do not conform to linear changes.
To address the limitations you pointed out, we have added a new paragraph in the discussion section (Section 5.4) (page 18, lines 533–537) to explain that the current model is more suitable for moderate temperature and humidity environments. The nonlinear effects under extreme conditions require further optimization through experiments, simulations under extreme conditions, or adaptive algorithms based on reinforcement learning.
We will focus our future research in this direction. Once again, thank you for your insightful comments, which are crucial for enhancing the rigor of our research.
Comment 4:While the experiments were carried out under controlled temperature conditions, in real-world applications of DFOS in underground caverns, temperature gradients are likely. A brief discussion on the expected influence of temperature cross-sensitivity and how it might be compensated in practice would improve the robustness of your conclusions.
Response 4: Response 4: Thank you for your questions. Regarding the impact of temperature gradients on the practical application of DFOS (Distributed Fiber Optic Sensing), we believe that temperature cross-sensitivity may significantly affect the measurement results in the following ways:
In underground caverns, temperature gradients can cause changes in the optical properties of the fiber optic cables, thereby affecting the measurement accuracy of the sensors. Variations in temperature may lead to changes in the refractive index of the fiber, which in turn can affect the propagation speed and intensity of the light signal, resulting in deviations in the measurement results. In a temperature-controlled underground space, the error caused by temperature effects can generally be kept very small.
To compensate for temperature cross-sensitivity in practice, the following methods can be adopted:(p9 line305-313)
Temperature Monitoring: Install temperature sensors at key locations to monitor environmental temperature changes in real time and correlate them with DFOS measurement results.
Data Correction: During the data processing stage, apply temperature compensation algorithms to correct the measurement results and eliminate the impact of temperature variations.
Multi-point Measurement: Add multiple temperature sensors in the fiber optic layout to obtain more comprehensive temperature distribution information, thereby improving the accuracy of compensation.
These measures can effectively reduce the impact of temperature gradients on DFOS measurement results, thereby enhancing the robustness of the conclusions. Our current experiment was conducted in a temperature-stable laboratory environment, providing a reference for measurement techniques in environments with minimal temperature differences. We will further discuss these compensation methods in future papers to enhance the understanding of the temperature sensitivity of DFOS in practical applications. Thank you for your valuable comments and for inspiring our research direction!
We will focus our future research in this direction. Once again, thank you for your insightful comments, which are crucial for enhancing the rigor of our research.
Minor Comment : - The introduction would be stronger if the authors explained why the cavern structures are needed.
- While the manuscript is generally understandable, the English writing contains several grammatical issues, redundant phrasing, and awkward sentence structures.
- Several figures suffer from low resolution or unclear line definitions, particularly Fig. 4 and Fig. 12. Labels within figures are often too small or blurry. Important labels and axis titles are difficult to read.
- Most figure captions are overly brief and lack sufficient explanation.
- A consistent visual style (e.g., font size, line thickness, marker types) in every graph should be improved.
- Typo: Braggrating (Row 32).
- The term “Distributed Fiber Optic Sensor (DFOS)” is introduced early on, but the full term is repeatedly used afterward instead of the defined abbreviation, which disrupts consistency.
- Line 6: Please remove the duplicated word “Correspondence: Correspondence:”.
Response : (1) Thank you to the reviewer for such detailed suggestions on this manuscript! We have revised the introduction to highlight the primary purpose of selecting the cavern structure and to enhance the professionalism of the introduction. (p1, line 25–46)
(2) Thank you for your feedback. We have proofread the manuscript based on your suggestions to correct grammatical issues, eliminate redundant wording, and improve sentence structure. Our goal is to ensure that the manuscript is expressed more clearly and smoothly to enhance readability and academic quality. Thank you for your valuable comments!
(3) Thank you for your feedback. We will reprocess Figures 4 and 12 to improve their resolution and line clarity. We will also increase the font size of labels and axis titles in the figure windows to ensure they are clearly readable. We will ensure that all important information is clearly presented in the figures to enhance the overall visualization. Thank you for your valuable comments!
(4) Thank you for your feedback. We will revise the figure captions to make them more detailed and explanatory so as to better convey the information presented in the figures. Our goal is to ensure that each caption clearly reflects the content and significance of the figure to improve the reader's understanding. Thank you for your valuable comments!
(5) Thank you for your suggestions. We have improved the visual style in each figure to ensure consistency, including the font size of illustrations, line thickness, and coordinate representation. This will help enhance the overall readability and professionalism of the figures and improve the reader's experience. Thank you for your valuable comments!
(6) Thank you for your review of each word. The revisions have been made. (p2, line 83)
(7) Thank you for your guidance on the professionalism of the paper. The revisions have been made to ensure consistency.
(8) Thank you for your careful review. The revisions have been completed.
